# CAC: Confidence-Aware Co-Training for Weakly Supervised Crack Segmentation

**DOI:** 10.3390/e26040328

**Published:** 2024-04-12

**Authors:** Fengjiao Liang, Qingyong Li, Xiaobao Li, Yang Liu, Wen Wang

**Affiliations:** 1Key Laboratory of Big Data Artificial Intelligence in Transportation (Beijing Jiaotong University), Ministry of Education, Beijing 100044, China; liangfj@bjtu.edu.cn (F.L.); liqy@bjtu.edu.cn (Q.L.); yliucit@bjtu.edu.cn (Y.L.); 2School of Computer Science and Technology, Jiangsu Normal University, Xuzhou 221116, China; 6020230055@jsnu.edu.cn

**Keywords:** weakly supervised learning, crack segmentation, co-training, confidence aware, pseudo-label dynamic division

## Abstract

Automatic crack segmentation plays an essential role in maintaining the structural health of buildings and infrastructure. Despite the success in fully supervised crack segmentation, the costly pixel-level annotation restricts its application, leading to increased exploration in weakly supervised crack segmentation (WSCS). However, WSCS methods inevitably bring in noisy pseudo-labels, which results in large fluctuations. To address this problem, we propose a novel confidence-aware co-training (CAC) framework for WSCS. This framework aims to iteratively refine pseudo-labels, facilitating the learning of a more robust segmentation model. Specifically, a co-training mechanism is designed and constructs two collaborative networks to learn uncertain crack pixels, from easy to hard. Moreover, the dynamic division strategy is designed to divide the pseudo-labels based on the crack confidence score. Among them, the high-confidence pseudo-labels are utilized to optimize the initialization parameters for the collaborative network, while low-confidence pseudo-labels enrich the diversity of crack samples. Extensive experiments conducted on the Crack500, DeepCrack, and CFD datasets demonstrate that the proposed CAC significantly outperforms other WSCS methods.

## 1. Introduction

Crack is one of the common defects in infrastructure such as roads, bridges, and tunnels. Regular maintenance contributes to effectively extending the service life of these infrastructures [1,2,3]. Automatic crack detection based on computer vision has become an instantly efficient and widely adopted method due to its non-contact and cost-effectiveness [4,5,6,7]. This task is essentially treated as a binary image segmentation problem, where each pixel in an image is classified as either “crack” or “non-crack”. Fully supervised crack image segmentation methods have demonstrated outstanding performance [8,9]. However, its effectiveness is contingent upon accurate pixel-level annotations. Pixel-level annotations are not only costly but also demand specialized expertise. To address this issue, weakly supervised crack segmentation (WSCS) attracts increasing attention [10]. The weakly supervised segmentation only needs coarse-grained information such as the bounding boxes [11], and patch-level labels [12,13,14]. Among them, the patch level labels greatly reduce annotation difficulty by only giving the category information of image patches, which are thus widely researched in the existing literature of WSCS.

To learn the accurate position and boundary information with merely the coarse-grained labels, the existing approach typically comprises two key stages: pseudo-label generation and segmentation model training. During the pseudo-labels generation stage, pixel-level pseudo-labels are derived from the crack detector using weak labels. During the segmentation model training stage, these pixel-level pseudo-labels are treated as ground truth for training the crack segmentation model. The majority of existing WSCS methods primarily concentrate on improving the accuracy of pixel pseudo-label generation. König et al. [12] introduced a WSCS method that produces high-quality pixel pseudo-labels by integrating class activation maps (CAMs) [15] with classifier localization and threshold segmentation. Similarly, Dong et al. [13] proposed a WSCS model rooted in patch-based techniques. They applied Conditional Random Field (CRF) post-processing to the crack CAM to generate initial crack pixel pseudo-labels, subsequently using these pseudo-labels to train the segmentation model. It is worth noting that both these methods rely exclusively on image class labels to generate pseudo-labels from CAMs, but CAMs often highlight the most discriminative part of the region and potentially compromise the overall integrity of crack regions. Al-Huda et al. [10] incorporated multi-scale CAMs as the initial crack pixel pseudo-labels. From the model training perspective, they introduced an Incremental Annotation Refinement (IAR) strategy, meticulously designed to iteratively improve the segmentation model. Moreover, the challenge arises from the inherent thin topology and low contrast of cracks, causing the activated crack pixels to appear coarser than the ground truth. Consequently, the generated pixel-level pseudo-labels contain a significant amount of noisy pixels, making them less reliable for model training. Nevertheless, this deep neural network model is sensitive to noisy labels, which can lead to model over-fitting in a mislabeled feature space [16].

To overcome the challenge of noisy labels in crack segmentation, the WSCS task can be defined as a robust learning problem of pixel-wise noisy labels. Notably, it can be observed that the center pixels of activated crack objects tend to have higher confidence scores in pseudo-labels, while the surrounding pixels have lower scores. This observation has inspired us to introduce a novel framework for co-training crack segmentation models, which leverages pixel pseudo-labels with varying confidence scores. In addition, we have incorporated a dynamic correction mechanism during the model training process to ensure that pixel-level pseudo-labels with different confidence scores are iteratively refined. Our approach aims to exploit the properties of noisy pseudo-labels and provide valuable insights for crack segmentation tasks.

The confidence-aware co-training (CAC) framework has been proposed to effectively address the issue of crack segmentation. Firstly, the CAC framework trains the crack classifier to generate class activation maps (CAMs) [15] using only patch-level class labels. These CAMs are subsequently refined to serve as initial crack pixel pseudo-labels. Secondly, the crack pixel pseudo-labels are categorized into two groups, high confidence and low confidence, based on the confidence scores within the CAM. Subsequently, a co-training crack segmentation model is introduced, where two collaborative networks are fed successively with both the high-confidence and low-confidence pseudo-label sample sets. The high-confidence pseudo-labels offer the model more reliable crack features and superior initialization parameters, while the low-confidence pseudo-labels provide a diverse range of crack features to enrich understanding. This strategy of co-training pseudo-labels with varying confidence scores harnesses the inherent characteristics of noisy pseudo-labels, enhancing the generalization of the model. Finally, to elevate the quality of the pseudo-labels, the intermediate results predicted by the segmentation model during the training process are used as weighting parameters to dynamically re-weight the pseudo-labels. This iterative optimization process ensures that both high-confidence and low-confidence pseudo-labels receive continual updates, ultimately contributing to a more accurate and robust segmentation model.

The contributions of this work are summarized as follows:A novel confidence-aware co-training framework is introduced for weakly supervised crack segmentation.Aiming at mitigating the effect of noisy pseudo-labels, a co-training mechanism is designed to iteratively refine the predicted pseudo-labels and accordingly learn a more robust crack segmentation model.A dynamic division strategy is proposed to handle the noisy pseudo-labels. Among them, the high-confidence pseudo-labels are utilized to optimize the initialization parameters and those with low-confidence enrich the diversity of crack samples.The effectiveness of the proposed CAC is demonstrated through extensive validation on three crack datasets: Crack500, DeepCrack, and CFD. The results showcase the superior performance of this approach compared to other state-of-the-art models.

The subsequent structure is as follows: Section 2 reviews existing work related to crack segmentation methods. Section 3 outlines the proposed CAC framework in detail. Section 4 presents the experimental setup and results pertaining to crack segmentation. Finally, Section 5 concludes the paper and offers insights for future research.

## 2. Related Works

This section provides an overview of crack segmentation research, covering both fully supervised and weakly supervised crack segmentation methods.

### 2.1. Fully Supervised Crack Segmentation Method

In recent years, deep neural network techniques have exhibited remarkable performance in the realm of image semantic segmentation [17]. These techniques have also found widespread application in the detection of cracks on infrastructure surfaces such as pavements and bridges [4]. Unlike natural images, cracks present a distinct linear topological structure with discontinuous, low-contrast crack pixels. Consequently, fully supervised crack segmentation methods have traditionally amalgamated principles of edge detection [18,19] with semantic segmentation frameworks. These methods fuse multi-scale features to distinguish cracks from the background. A noteworthy example is DeepCrack [20], which integrates multi-level features within the fully convolutional networks (FCNs) for semantic segmentation [21] to achieve superior crack recognition. Zou et al. [22] introduced DeepCrack, which is a crack segmentation framework that incorporates multi-scale features of encoder and decoder in the SegNet for image segmentation [23] to capture crack structures. Both of these approaches propose the strategy of multi-scale feature fusion, which effectively addresses the challenge of crack continuity.

Building upon multi-scale fusion, several methods introduce an attention mechanism to emphasize crack pixels within the segmentation. Chen et al. [24] proposed a crack segmentation method that integrates an attention mechanism into the U-Net [25] framework to efficiently focus on crack pixel information. Sun et al. [26] introduced a pavement crack segmentation framework. This framework incorporates a novel multi-scale attention module within the decoder of DeepLabv3+ for semantic segmentation [27]. The primary function of this multi-scale attention module is to generate an attention mask. This mask dynamically assigns weights to features in different layers, thereby enhancing multi-scale feature fusion. Wang et al. [28] developed a saliency detection method, RENet, for pavement cracks. It incorporates a rectangular convolutional pyramid module to fuse contrast information between crack and background at different scales. Furthermore, the method introduces a crack edge enhancement network to filter out background noise and refine crack boundaries globally.

The advent of Transformer [29] has brought about a paradigm shift in vision tasks. In the field of crack segmentation [30,31,32], Transformer overcomes the limitation of fully exploiting contextual information in convolutional neural networks (CNNs) by capturing long-range dependencies in images through a self-attentive mechanism [33]. Presently, fully supervised crack segmentation methods that rely on data-driven approaches have achieved optimal performance in the field. However, these methods are contingent on high-precision manual annotation, incurring substantial costs.

### 2.2. Weakly Supervised Crack Segmentation Methods

To mitigate the cost of pixel-level annotation, weakly supervised crack segmentation (WSCS) methods have garnered increasing interest [34,35,36,37]. Presently, common weak labels in WSCS methods encompass category labels, bounding boxes, scribble lines, and other variants. Zhang et al. [11] introduced a WSCS method that leverages the bounding box labels of cracks to train an object detection network for the generation of initial pseudo-labels. These initial pseudo-labels are subsequently refined into precise pixel-level crack labels through the application of region-growing and GrabCut algorithms [38]. Zhang et al. [39] developed CrackGAN, a crack segmentation network based on generative adversarial networks [40] using scribble lines as weak labels. This method employs an asymmetric U-shape generator to address the “all-black” problem caused by the imbalance between crack and background pixels.

Among the various weak labels mentioned, patch-level category labels prove to be the most accessible and widely adopted. Fan et al. [41] presented a patch-based crack segmentation method. It combines traditional image processing methods with a deep classification model. The deep classification model is responsible for locating the cracks, while traditional image processing techniques are employed for bilateral filtering and threshold segmentation on the crack image patches. Nevertheless, the prevalent WSCS method, which relies exclusively on image-level category labels, follows a two-stage approach. This process entails the generation of crack pixel labels and the subsequent training of the segmentation model. Image-level category labels are utilized to train the crack classifier, generating CAMs [15] as pixel-level crack pseudo-labels. These pseudo-labels are then used to train the crack segmentation model. König et al. [12] proposed a WSCS method that employed location with a classifier and threshold segmentation to generate crack pixel-level pseudo-labels. These pseudo-labels serve as training data for the crack segmentation model. This approach enables the approximate identification of crack locations by the crack classifier while mitigating the impact of noisy background pixels. However, this method may overlook tiny cracks. Dong et al. [13] introduced a weakly supervised patch-based crack segmentation method. It obtains CAMs as the initial pixel pseudo-labels from a trained classification network and subsequently refines these labels using CRF. Wang et al. [42] developed Crack-CAM, a pixel-level WSCS method that incorporates clustering within the CNN classifier to enhance crack features and improve the quality of the pseudo-labels assigned to crack pixels.

While several WSCS methods have prioritized the generation of higher-quality crack pixel pseudo-labels, the most recent approaches have extended their focus to enhancing the training process of the segmentation model. Al-Huda et al. [10] proposed a weakly supervised pavement crack segmentation method. This approach employs the strategy of multi-scale CAM fusion to enhance the completeness of pseudo-labels and improve the quality of the initial pseudo-labels. Furthermore, the method incorporates an incremental annotation refinement (IAR) module to progressively enhance the pseudo-labels and iteratively optimize the crack segmentation model. Al-Huda et al. [14] also introduced a hybrid deep learning approach for WSCS. This method combines CAMs from the CNN classifier with encoder-extracted features. These fused features are then input into the decoder of the segmentation model to improve the quality of crack segmentation. However, our proposed approach not only concentrates on enhancing pseudo-labels during the model training process but also capitalizes on the inherent characteristics of noisy pseudo-labels. This approach allows the crack segmentation model to be iteratively optimized for better performance.

## 3. Methods

### 3.1. Overview

This paper proposes a novel method named confidence-aware co-training (CAC) for weakly supervised crack segmentation. The whole framework of CAC is shown in Figure 1. As aforementioned, CAC is designed to iteratively refine pixel-level pseudo-labels by co-training data with high confidence and low confidence. The proposed CAC method consists of three main modules: (a) crack pseudo-labels generation, (b) dynamic division of confidence pseudo-labels, and (c) co-training of segmentation models.

The first module, the crack pseudo-label generation module, involves training a crack image classifier using solely patch-level class labels to generate crack class activation maps (CAMs), serving as initial crack pseudo-labels. During the dynamic refining process of the pixel-level pseudo-labels, the confidence measures of the pseudo-labels also change accordingly. Among them, the pixels with high-confidence pseudo-labels are utilized to mine intrinsic discriminative patterns, while those with low-confidence pseudo-labels carry some partially activated crack pixels and boundary information. This motivates the idea of confidence-aware co-training to enable a more robust crack segmentation model. Specifically, in the second stage, the crack pseudo-label is divided into two sets of pseudo-labels, each assigning different confidence scores. In the third stage, two collaborative crack segmentation networks are co-trained by both the high-confidence and low-confidence pseudo-label sample sets, leading to the optimization of the pseudo-label quality during the training process. These main modules are respectively detailed in the following three subsections.

### 3.2. Crack Pseudo-Label Generation

The CAM method proposed as a visualization tool for convolutional neural networks is widely applied to weakly supervised image segmentation tasks [43,44,45]. Image-level labels are employed to train an image classification model, and the CAM method generates pixel-level pseudo-labels based on up-sampling the regions of interest of the classification model. The CAM operates by applying global average pooling (GAP) to the final convolutional layer features in the classifier and then reversely mapping the weight of the GAP output layer back to the output feature layer. Building on the CAM method, the Grad-CAM method [46] omits the GAP layer and directly propagates the gradient information of the target object into the final convolutional layer, thereby providing a more direct focus on the target pixels. Following this principle, this work employs the ResNet network [47] to train a classification model and applies the Grad-CAM method to generate an initial pseudo-label for each pixel. Since crack pixels typically represent a relatively small proportion of the entire image and tend to be overlooked during feature extraction due to their linear topology, this work constructs an image patch dataset with class labels (crack or non-crack) for training the classification model. The Grad-CAM method based on local image patches aims to maximize the activation of the crack area within the entire image.

Given an image *X*, it is divided into a crack image patch $x=\left\{x_{i}|i=1,2,\ldots ,n\right\}$ with overlapping regions using a step size of *d*. Each crack image patch $x_{i}$ is fed into the classification model, resulting in the generation of the corresponding CAM $s_{i}$. These $s_{i}$ are then synthesized to produce the crack CAM *S* corresponding to the original image *X*. The crack pseudo-labels are generated using two distinct methods for comparison. The first method employs location information with the crack classifier and threshold segmentation as proposed by König et al. [12]. The second method utilizes CRF post-processing to refine the crack CAMs, following the approach by Dong et al. [13]. Both of these methods produce pseudo-labels with varying degrees of quality. According to Equation (Equation 1), the initial pseudo-label *M* is formulated as follows: if $P_{ij}$ is greater than $\tau_{1}$, then $M_{ij}$ is set to $P_{ij}$, and otherwise, $M_{ij}$ is set to 0:(1)$$M_{ij}=\left\{\begin{array}{cc} P_{ij}, & \;P_{ij}\geq \tau_{1};\\ 0, & \mathrm{otherwise}. \end{array}\right.$$Here, $P_{ij}$ denotes the pixel values of the *i*-th row and *j*-th column in the pseudo-label image *P*. The threshold $\tau_{1}$ corresponds to the value selected from the top *k* elements in the histogram of pseudo-label *P*. Based on the crack pseudo-label pixels statistics, *k* is set to a larger value of 15% to retain as many crack pixels as possible. This thresholding process is used to create the gray-scale image *M*, which serves as the initial pseudo-label for crack segmentation.

### 3.3. Dynamic Division of Confidence Pseudo-Labels

This section addresses the issue of noisy pseudo-labels generated by CAM methods and focuses on utilizing these labels to train a robust crack segmentation model. Since CAM methods tend to activate only the most discriminative regions of the crack, the proposed method divides the noisy pseudo-labels into two sets: high-confidence and low-confidence pseudo-labels. These two sets are used to co-train the segmentation network.

In the previous stage, the initial noisy pseudo-labels are denoted as *M*, where $M_{ij}\in \left[0,255\right]$. At the current stage, these *M* are divided into high-confidence pseudo-labels *H* and low-confidence pseudo-labels *L* by threshold processing. The division of pseudo-labels is described as follows:(2)$$H_{ij}=\left\{\begin{array}{cc} 255, & M_{ij}\geq \tau_{2};\\ 0, & \;\mathrm{otherwise}.\; \end{array}\right.$$
(3)$$L_{ij}=\left\{\begin{array}{cc} 255, & \tau_{1}\leq M_{ij}<\tau_{2};\\ 0, & \mathrm{otherwise}. \end{array}\right.$$Here, $\tau_{1}$ is defined in Equation (Equation 1), and $\tau_{2}$ is the threshold corresponding to the top C% in the gray-scale histogram of *M*. More details about the configuration of *C* can be found in Section 4.9. The high-confidence sample set $D_{H}$ provides accurate supervision for the segmentation network, while the low-confidence sample set $D_{L}$ offers richer information to the network. During the training process, the semantic information of uncertain crack pixels is corrected by dynamically updating the noisy pseudo-labels *M*. The correction weights are defined as α, the size of α is the same as that of *M*, $\alpha_{ij}\in \left[0,1\right]$, and α is initialized as a unit matrix. During the model training process, after each training epoch, the result map output from the segmentation model is used as the correction weights α and represents the probability value of each pixel belonging to the crack in the input image. The correction process for the pseudo label *M* is described as:(4)$$M_{t+1}=\alpha \cdot M_{t}.$$This approach progressively refines the pixel pseudo-labels as the model is iteratively optimized.

### 3.4. Co-Training of Segmentation Models

Instead of directly training a segmentation model using noisy labels, the CAC framework is inspired by the optimization methodology of Model-Agnostic Meta-Learning (MAML) [48] to overcome the challenges posed by noisy pseudo-labels. Although our problem setting differs significantly from MAML, we are inspired by the optimization methodology of MAML and propose a co-training strategy. The CAC framework employs two collaborative networks that share the same architecture. This design allows high-confidence and low-confidence pseudo-labels to co-train the segmentation model effectively. High-confidence pseudo-labels offer accurate supervision information, allowing for better initialization parameters for the segmentation model. In contrast, low-confidence pseudo-labels contain some partially activated crack pixels and background pixels around the cracks, providing richer data features.

In the CAC framework, after collecting the required image and pseudo-labels in the previous stage, the semantic segmentation dataset $D={\left\{\left(X^{i},H^{i},L^{i}\right)\right\}}_{i=1}^{N}$ is obtained, and $X^{i},H^{i},L^{i}$ represent the original image, high-confidence pseudo-labels, and low-confidence pseudo-labels, respectively. Two collaborative networks, denoted as $f_{\theta }$ and $f_{\theta^{\prime }}$, are constructed. The parameters of the two networks are θ and $\theta^{\prime }$, respectively. The details of the co-training strategy are shown in Figure 2.

Firstly, the high-confidence pseudo-label samples set $D_{H}={\left\{\left(X^{i},H^{i}\right)\right\}}_{i=1}^{N}$ is fed into the crack segmentation network $f_{\theta }$, resulting in prediction results ${P_{1}}^{i}$, where $P_{1}^{i}=f_{\theta }\left(X^{i}\right)$. The segmentation loss $L_{1}\left(P_{1}^{i},H^{i}\right)$ is calculated using the focal loss [49]. $L_{1}$ is defined as:(5)$$\begin{array}{c} L_{1}\left(P_{1}^{i},H^{i}\right)=-\sum_{i=1}^{N}\beta {\left(1-P_{1}^{i}\right)}^{\gamma }H^{i}\log P_{1}^{i}+\left(1-\beta \right){\left(P_{1}^{i}\right)}^{\gamma }\left(1-H^{i}\right)\log\left(1-P_{1}^{i}\right). \end{array}$$Here, low-confidence crack pixels are not involved in training $f_{\theta }$ to avoid the interference of low-confidence pixels, as background pixels, with the training of $f_{\theta }$. So $P_{1}^{i}$ and $H^{i}$ are masked to hide the low-confidence crack pixels portion of pseudo-labels as shown in Figure 3a. Moreover, β is used to balance the number of positive and negative samples, and γ regulates the imbalance between easy-to-discriminate and hard-to-discriminate samples. A one-step gradient update $\theta^{\prime }$ is performed: $\theta^{\prime }\leftarrow \theta -\nabla_{\theta }L_{1}$.

Then, the low-confidence pseudo-label samples set $D_{L}={\left\{\left(X^{i},L^{i}\right)\right\}}_{i=1}^{N}$ is fed into the crack segmentation network $f_{\theta^{\prime }}$, resulting in prediction results ${P_{2}}^{i}$, where $P_{2}^{i}=f_{\theta^{\prime }}\left(X^{i}\right)$. The segmentation loss $L_{2}\left(P_{2}^{i},L^{i}\right)$ is calculated as:(6)$$\begin{array}{c} L_{2}\left(P_{2}^{i},L^{i}\right)=-\sum_{i=1}^{N}\beta {\left(1-P_{2}^{i}\right)}^{\gamma }L^{i}\log P_{2}^{i}+\left(1-\beta \right){\left(P_{2}^{i}\right)}^{\gamma }\left(1-L^{i}\right)\log\left(1-P_{2}^{i}\right). \end{array}$$Here, $P_{2}^{i}$ and $L^{i}$ are masked to hide the high-confidence crack pixel portion as shown in Figure 3b. High-confidence crack pixels are not involved in training $f_{\theta^{\prime }}$ to avoid the high-confidence pixels, as background pixels, interfering with the training of $f_{\theta^{\prime }}$.

Finally, the total loss $L_{total}$ is calculated and is formulated as:(7)$$L_{\mathrm{total}}=\lambda L_{1}+\left(1-\lambda \right)L_{2}.$$Here, λ is a parameter used to adjust the balance of the contribution of high-confidence and low-confidence sample sets. The gradient $\theta \leftarrow \theta -\nabla_{\theta }L_{total}$ is updated for the segmentation network $f_{\theta }$, which is used as the final test model. This process ensures that the model learns from both high-confidence and low-confidence pseudo-labels, improving the robustness of the segmentation model while exploiting the properties of the noisy pseudo-labels.

Algorithm 1 describes the overall flow of the proposed CAC method.
**Algorithm 1:** CAC algorithm.
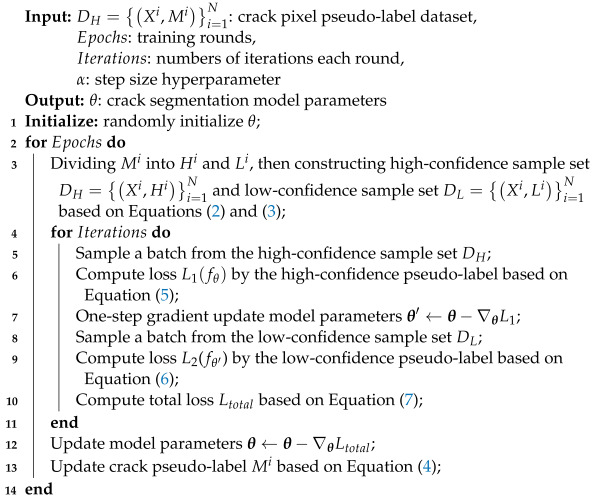


## 4. Experimental Results, Comparisons, and Analysis

This section discusses the implementation details of the proposed CAC and compares its effectiveness to other state-of-the-art WSCS methods presented in recent years using three crack image datasets.

### 4.1. Datasets

Following the literature [12], the Crack500 dataset [50] is employed to train the classification network for computing grad-CAM and generating initial pseudo-labels. This dataset consists of 1896 crack images, each with a resolution of 648 × 484 pixels. To augment the data, these images are sliced into patches of 128 × 128 pixels, and augmentation techniques like rotation and flipping are applied. In total, 556,448 images are used for training the classification network. The dataset contains 238,820 images with cracks and 317,628 images without cracks.

The crack segmentation network is trained on the original images and the generated pseudo-labels for crack pixels in the Crack500 training set. The segmentation model is then tested on three different datasets:Crack500 testing dataset [50]: This dataset consists of 1124 crack images. Crack500 is a pavement cracking dataset that is collected with a mobile phone on the campus of Temple University.CFD dataset [51]: It contains 118 crack images, each with a resolution of 320 × 480 pixels, which reflect urban road surface conditions in Beijing, China. This dataset includes various types of noise such as shadows, oil spots, and water stains.DeepCrack dataset [20]: This dataset comprises a total of 537 images, each with a resolution of 544 × 384 pixels. It includes crack data with multiple textures, scenes, and scales.

These datasets serve as the foundation for training and testing the CAC framework. The diversity in dataset sources and characteristics allows for a comprehensive evaluation of the performance and its ability to handle different types of crack data.

### 4.2. Evaluation Metrics

Following the original literature [12,22], three different F1-based metrics are employed for the evaluation of the proposed CAC: optimal dataset scale (ODS), optimal image scale (OIS), and average precision (AP). The ODS represents the best F1 scores on the entire dataset for a fixed threshold. The OIS denotes the aggregate F1 scores calculated for each image in the dataset using the best threshold for each image. The AP denotes the area under the precision–recall curve [52]. ODS and OIS are represented as follows:(8)$$Pr=\frac{TP}{TP+FN},$$
(9)$$Re=\frac{TP}{TP+FP},$$
(10)$$F1=2\times \frac{Re\times Pr}{Re+Pr},$$
(11)$$OIS=\frac{1}{N}\sum_{i=1}^{N}\max\left\{F1_{t}^{i}:\forall t\in \left\{0.01,0.02,\ldots ,0.99\right\}\right\},$$
(12)$$ODS=\max\left\{\left\{\frac{1}{N}\sum_{i=1}^{N}F1_{t}^{i}\right\}:\forall t\in \left\{0.01,0.02,\ldots ,0.99\right\}\right\}.$$Here, Pr, Re, F1, TP, FP, TN, and FN denote precision, recall, *F*1-score, true positive, false positive, true negative, and false negative, sequentially. AP is represented as follows:(13)$$AP=\sum_{t=0.01}^{t=1}\frac{1}{T}\left(Re_{t}-Re_{t-0.01}\right)Pr_{t},$$where *t* is the selected threshold and *T* is set to 100.

### 4.3. Implementation Details

#### 4.3.1. Environment

Our experiments are conducted on a deep learning workstation running Ubuntu 16.04 LTS containing a CPU of Intel(R) Xeon(R) CPU E5-2695 v4 @ 2.10 GHz and a Nvidia Titan XP GPU with 8 GB of RAM. The framework used for the experiments is pytorch1.2.

#### 4.3.2. Experimental Setting

This section provides detailed information about the experimental settings for the crack pixel-level pseudo-label generation and the training and testing phases of the crack segmentation model. During the crack pixel-level pseudo-label generation, a binary classification model based on ResNet50 architecture is used for generating pixel-level pseudo-labels for crack images. The classification model is trained for 10 epochs with a batch size of 16. The initial learning rate is set to $1\times 10^{-3}$, and the learning rate is reduced by a factor of 10 per epoch. Stochastic Gradient Descent (SGD) is employed as the optimizer with a momentum value of 0.9. Grad-CAM is generated for crack images by training the binary classification network (ResNet50). Two different processes are used to obtain the initial pseudo-label from the Grad-CAM. One process involves fusing the CAMs and crack locations obtained from the classification network, followed by threshold segmentation to generate the initial crack pseudo-labels [12]. The other process refines the CAMs using CRF post-processing to produce initial crack pseudo-labels [13]. During the division phase of different confidence pseudo-labels, a threshold segmentation method P-Tile [53] is employed due to its anti-noise capabilities compared to other threshold segmentation methods. P-Tile adapts the threshold dynamically based on the gray-scale histogram statistics. The division threshold values $\tau_{1}$ and $\tau_{2}$ are selected based on prior information from DeepCrack [22] to match the statistics of crack pixels. During the phase of crack pseudo-label generation, *k* is set to 15% by P-Tile.

During the training phase of the crack segmentation model, DeepCrack, a specific segmentation framework, is chosen for training the crack segmentation model. The model is trained on the Crack500 training set, which includes original images and pseudo-labels of crack pixels with different confidence scores. Training takes place for 30 epochs with a batch size of 4. The initial learning rate is set to $1\times 10^{-3}$ and is reduced by a factor of 10 every 10 epochs. The used optimizer is SGD with a momentum value of 0.9. The input image size for DeepCrack is set to 256 × 256 pixels. Focal loss, a type of binary cross-entropy loss, is employed to calculate the error between the predicted output and the corresponding pseudo-labels. During segmentation model testing, three datasets including the Crack500 test set, CFD, and DeepCrack are involved in the inference process.

### 4.4. Evaluation on Crack500

As shown in Table 1, the results of the Crack500 testing dataset demonstrate the effectiveness of the proposed CAC method compared to the state-of-the-art methods. Compared to using CAM as pseudo-labels for direct segmentation model training, CAC achieves improvements in terms of the following metrics: ODS increases by 0.76% to reach a value of 53.88%; OIS increases by 1.58% to reach a value of 58.44%; and AP improves by 7.18% to reach a value of 57.07%. Compared to PWSV methods, refining CAM with CRF post-processing as pseudo-labels for direct segmentation model training [13], CAC achieves improvements in the following metrics: ODS increases by 4.68% to reach a value of 61.22%; OIS improves slightly, with a 0.34% increase to reach a value of 64.07%; and the AP value is closer, with a value of 65.10%. Compared to the GPLL method that employed location with a classifier and thresholding as the initial crack pseudo-labels for direct segmentation model training [12], CAC achieves significant improvements in the following metrics: ODS increases substantially by 15.39% to reach a value of 60.43%; OIS improves by 7.81%, reaching a value of 64.50%; and AP sees a remarkable improvement of 18.19% to reach a value of 63.65%. It is important to note that the fully supervised method (FSV), which uses pixel labels with precise annotations, serves as an upper bound on the performance of the segmentation model. These results indicate that CAC outperforms other methods when it comes to optimizing the training of models. The model is more resistant to noise interference, resulting in a refined crack segmentation map with less noise and more visible cracks, particularly with a higher recall of crack pixels as demonstrated in Figure 4.

### 4.5. Evaluation on CFD

As shown in Table 2, the evaluation of the CFD dataset demonstrates the performance of CAC compared to other state-of-the-art methods. The CAC performs better than existing methods when using pseudo-labels of the same quality. Specifically, when GPLL is used as the initial pseudo label [12], CAC achieves the best segmentation results, which hold on an ODS value of 25.31%, an OIS value of 31.55%, and an AP value of 18.55%. However, when using CAMs as pseudo-labels, the pseudo-label refinement strategy is less effective due to the roughness of the intermediate results α. Compared to the Crack500 dataset, the cracks in the CFD dataset are thinner and more challenging to detect, resulting in lower performance for the segmentation results. Note that the FSV in Table 2 refers to the segmentation model trained on Crack500 rather than CFD, and it does not show the best performance because of its poor generalization ability. Similar results can be observed in Table 3. The visualizations provided in Figure 5 indicate that CAC has less noise compared to other methods, but it may still miss thinner cracks. These results suggest that CAC outperforms other methods on the CFD dataset in terms of handling noisy pseudo-labels, although it still faces challenges with very thin cracks. The exploration of trying to use the output results of different layers (shallow feature or deep feature) in the DeepCrack framework is helpful to further enhance the pseudo-label correction strategy.

### 4.6. Evaluation on DeepCrack

As shown in Table 3, the evaluation of the DeepCrack dataset shows that CAC achieves superior performance compared to other state-of-the-art methods. The CAC outperforms other methods on the DeepCrack dataset when using the same quality of pseudo-labels. Specifically, when GPLL is used as the initial pseudo label [12], the CAC achieves state-of-the-art segmentation results, which hold on an ODS value of 71.01%, an OIS value of 77.98%, and an AP value of 75.51%. In Figure 6, the visualized crack segmentation results of CAC and existing methods on the DeepCrack dataset with three different pseudo-labels demonstrate that CAC produces segmentation results with fewer noisy pixels compared to other methods. The CAC displays excellent generalization performance on the DeepCrack dataset, showcasing its robustness and superior performance. These results highlight that CAC is highly effective in handling noisy pseudo-labels and demonstrates excellent generalization performance, making it a state-of-the-art method for crack segmentation on the DeepCrack dataset.

### 4.7. Model Performance Discussion and Summary

From our experimental results, we observed that the performance of segmentation results significantly improves when the segmentation model is trained using crack pseudo-labels of comparable quality. Surprisingly, the noise inherent in these crack pseudo-labels has minimal impact on the performance of our model. Consequently, our approach demonstrates remarkable resilience to noise interference from pseudo-labels. The segmentation results on the CFD dataset are relatively poor compared to the Crack500 dataset. This can be attributed to the thinner and lower-contrast nature of the cracks in the CFD dataset, which can be more challenging to detect and segment accurately. The DeepCrack dataset, with a data distribution more similar to the crack distribution in the training set, yields better segmentation results with CAC.

The results presented in Figure 7 and the provided explanation indicate that the proposed CAC method faces challenges in detecting thin and low-contrast cracks. All WSCS methods for detecting thin cracks, including our proposed method, may experience instances of missed detection.

The experimental results presented in Table 2 and Table 3 reveal that the weakly supervised crack segmentation method exhibits superior generalization performance compared to the fully supervised method on both the CFD and DeepCrack datasets. Analyzing the training data perspective, the fully supervised approach in this study utilizes the Crack500 dataset, potentially leading to model over-fitting. In contrast, the weakly supervised approach incorporates pseudo-labels with varying degrees of noise, which contributes to enhanced model generalization. Consequently, the proposed CAC offers a balancing mechanism that effectively mitigates the influence of dataset-specific features and noise during the data-fitting process.

### 4.8. Ablation Experiments

Table 4 presents the performance of different modules in the CAC framework using the pseudo-labels generated by the fusion of CAM and crack location [12] on the Crack500. The first row shows the segmentation performance of directly training the segmentation model using the initial pseudo-labels. This is the baseline method. The second row demonstrates that the co-training module provides a significant improvement in the segmentation performance of the model. Co-training with high-confidence and low-confidence pseudo-labels allows the model to better leverage the features of noisy labels, resulting in enhanced segmentation. The third row showcases the performance of the dynamical division module for pseudo-labels with different confidence scores. The results indicate that as the quality of pseudo-labels improves, the segmentation performance also improves. In summary, the experimental results suggest that the co-training module and the dynamical division module play crucial roles in enhancing the segmentation performance of the CAC model. These modules help the model make better use of pseudo-labels, leading to improved results.

Figure 8 visually demonstrates the pixel pseudo-label correction during model training. It can be seen that the crack CAM is used as the initial coarse pseudo-label Iter_0, which contains a significant amount of noise. As the iterative number increases, it shows steady trends, and the pseudo-label is refined gradually. This supports the motivation of confidence-aware co-training experimentally and illustrates that the proposed CAC can effectively strengthen the ability to resist noises.

### 4.9. Parameter Experiments

Figure 9 illustrates the impact of different values of the parameter λ on the total loss of the segmentation model using the pseudo-labels generated by the fusion of CAM and crack location [12] on the Crack500. The graph shows that the crack segmentation model performs optimally when λ is set to 0.7. In this configuration, the contribution of the high-confidence sample set accounts for 70% of the total loss, while the contribution of the low-confidence sample set accounts for the remaining 30%. This balance between high-confidence and low-confidence sample sets yields the best segmentation results. As λ varies, the performance of the segmentation model changes. For values below 0.7, the model appears to under-utilize the high-confidence sample set, leading to sub-optimal performance. Conversely, for values above 0.7, the model starts to favor the high-confidence samples to a greater extent, which can also negatively affect the overall performance. The figure indicates that as the parameter λ changes, the performance of the model fluctuates within a relatively small range (around 10%).

Figure 10 illustrates the impact of different division ratios, parameter *C*, on the performance of the crack segmentation model when using pseudo-labels generated by the fusion of CAM and crack location on the Crack500. It shows that the performance of the model is optimized when *C* is set to 5, which means that the top 5% of the initial noisy pseudo-labels are divided as a high-confidence sample set, and the remaining pixels are divided as a low-confidence sample set. These results indicate that selecting an appropriate division ratio is crucial for achieving the best segmentation performance with the CAC model.

## 5. Conclusions and Future Work

We contributed an effective strategy named confidence-aware co-training (CAC) for weakly supervised crack segmentation tasks. The proposed CAC employed the co-training of crack segmentation networks with pseudo-labels under different degrees of confidence to mitigate the impact of noisy labels and enhance pseudo-label accuracy. The approach leveraged high-confidence pseudo-labels for better initialization parameters and reduced over-fitting to noisy pixels while using low-confidence pseudo-labels to provide richer features for the model. Pseudo-labels were dynamically updated to improve their quality during model training by iterative optimization model. The experimental results have demonstrated the superiority of the proposed CAC approach over state-of-the-art methods on the Crack500, CFD, and DeepCrack datasets. The CAC framework offers a promising approach to weakly supervised crack segmentation and highlights the potential for further improvements in crack segmentation methodologies. For future work, more exploration will be performed to meet the challenges of CAC in preserving the connectivity of cracks.

## Figures and Tables

**Figure 1 entropy-26-00328-f001:**
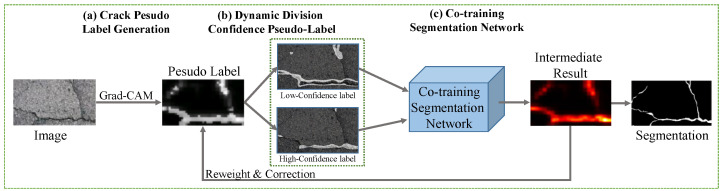
Overview of the proposed CAC framework. (**a**) Pixel-level crack pseudo-labels are produced using the patch-based Grad-CAM method. (**b**) The prediction results generated by the segmentation model at each training epoch are utilized as weights for updating the pseudo-labels. The pseudo-labels are dynamically divided into high-confidence and low-confidence sets based on the confidence scores associated with the crack pixels. (**c**) A co-training network is designed to train the segmentation model by collaborating high-confidence pseudo-label samples with low-confidence ones. The purpose is to train the segmentation model progressively and iteratively optimize its performance.

**Figure 2 entropy-26-00328-f002:**
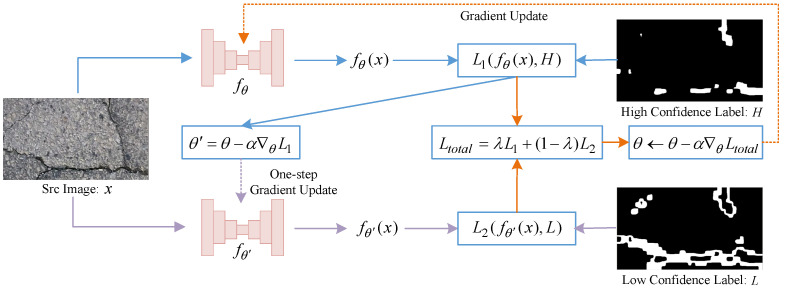
Co-training segmentation networks with pseudo-labels of different confidence.

**Figure 3 entropy-26-00328-f003:**
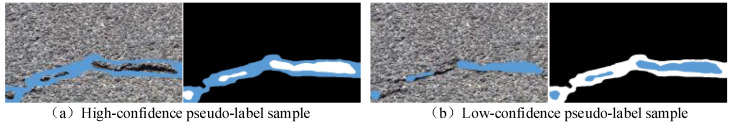
The graphic shows the high-confidence and low-confidence pseudo-label cracking samples. The blue parts are the masked pixels at the current confidence level.

**Figure 4 entropy-26-00328-f004:**
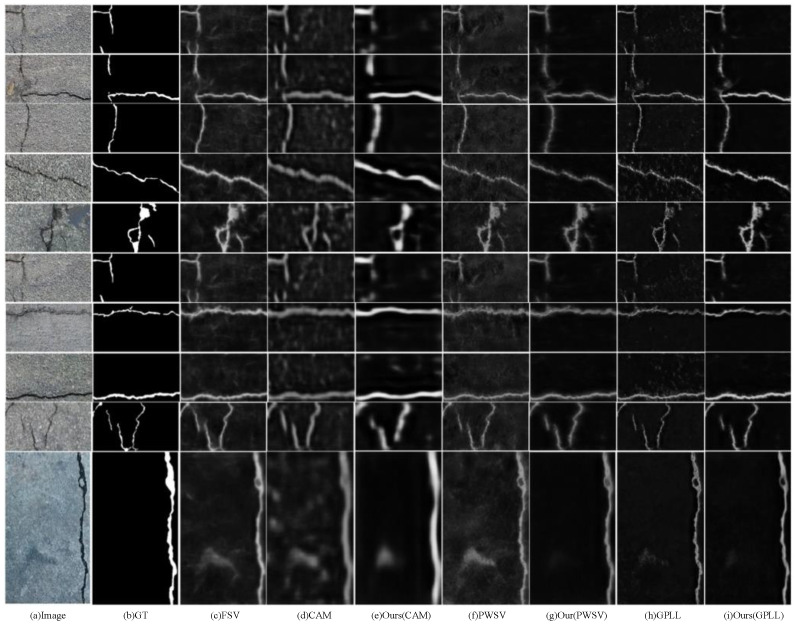
Visualization of the crack segmentation results of the different methods on the Crack500 testing dataset.

**Figure 5 entropy-26-00328-f005:**
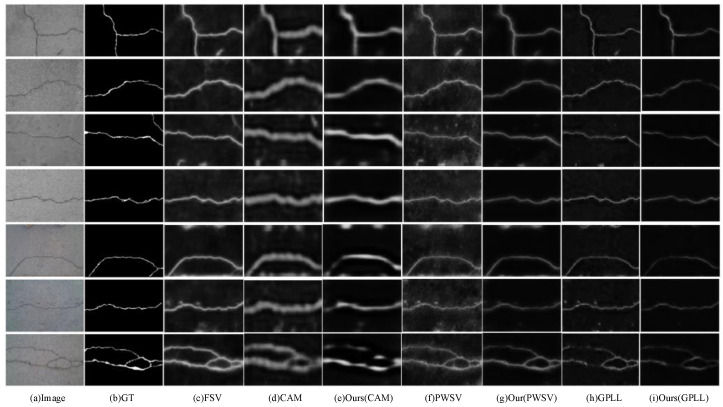
Visualization of the crack segmentation results of the different methods on the CFD dataset.

**Figure 6 entropy-26-00328-f006:**
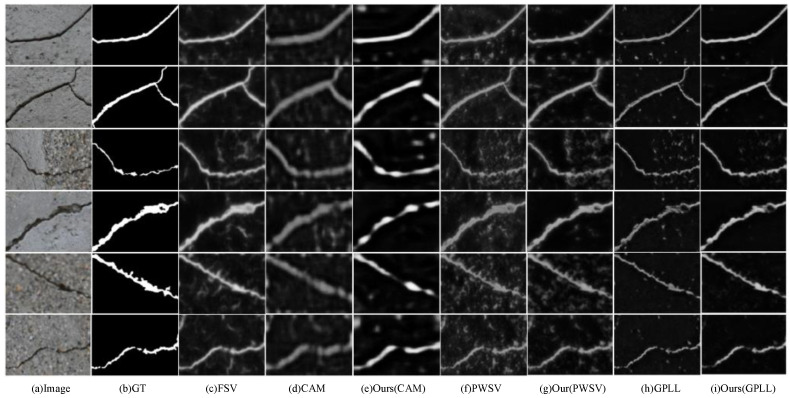
Visualization of the crack segmentation results of the different methods on the DeepCrack dataset.

**Figure 7 entropy-26-00328-f007:**
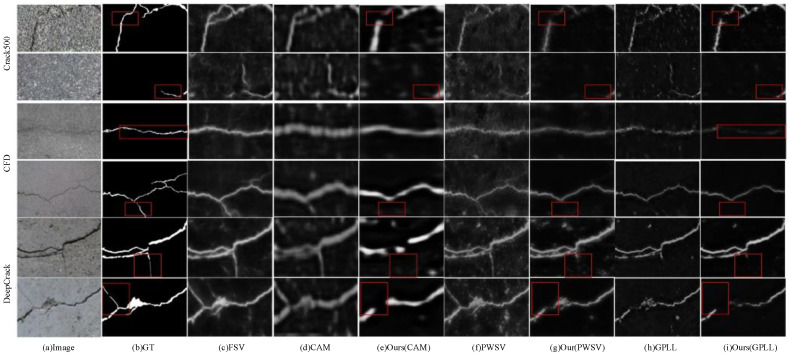
The graphic shows some example images with poor segmentation results under different datasets. The red box is the region where the detection error occurs.

**Figure 8 entropy-26-00328-f008:**
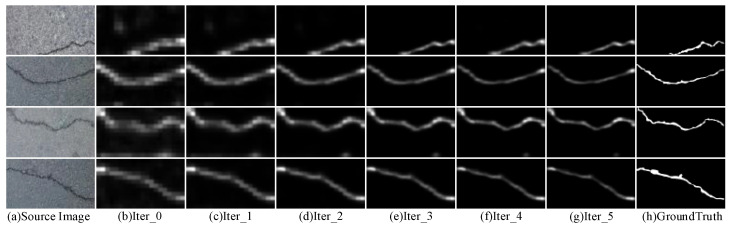
Visualization of the process of CAM pseudo-label correction during model iterations.

**Figure 9 entropy-26-00328-f009:**
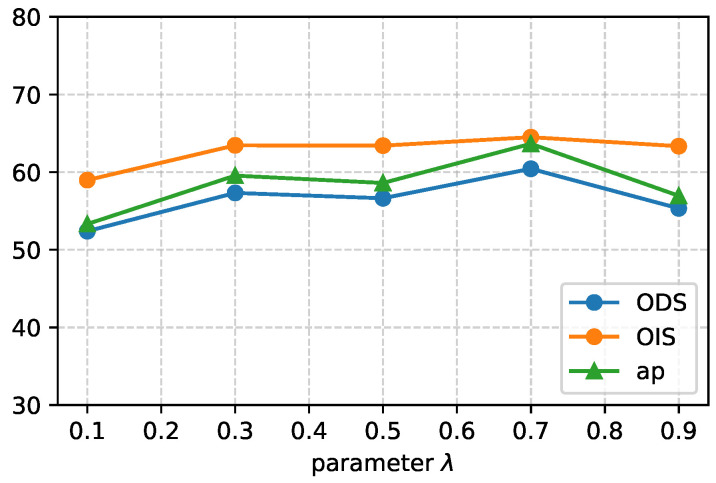
Performance of different λ in total loss for segmentation model on Crack500 testing dataset.

**Figure 10 entropy-26-00328-f010:**
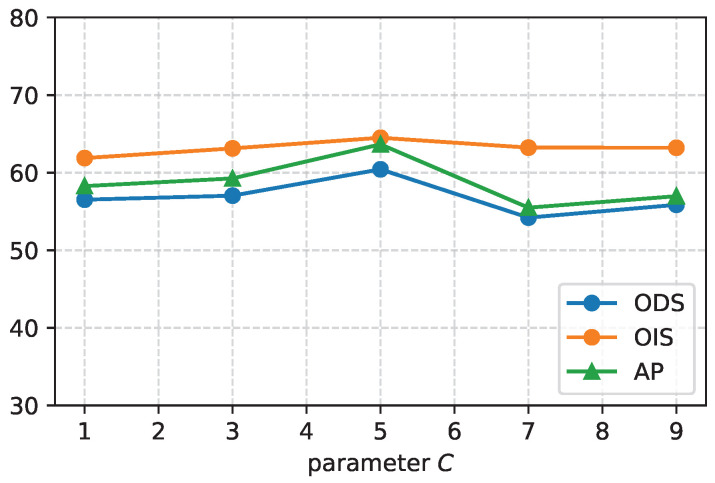
Performance of different division ratios *C* for the segmentation model on the Crack500 testing dataset.

**Table 1 entropy-26-00328-t001:** Evaluation of the segmentation results ODS, OIS, and AP of different methods on the Crack500 test set (%).

Methods	ODS	OIS	AP
FSV	66.20	71.97	76.70
CAM [46]	53.12	56.86	49.89
PWSV [13]	56.54	63.73	65.13
GPLL [12]	45.04	56.69	45.46
Ours (CAM)	53.88	58.44	57.07
Ours (PWSV)	**61.22**	64.07	**65.10**
Ours (GPLL)	60.43	**64.50**	63.65

**Table 2 entropy-26-00328-t002:** Evaluation of the segmentation results ODS, OIS, and AP of different methods on the CFD dataset (%).

Methods	ODS	OIS	AP
FSV	16.67	24.35	6.31
CAM [46]	23.16	17.52	14.07
PWSV [13]	8.56	14.46	7.72
GPLL [12]	18.74	19.41	14.88
Ours (CAM)	22.87	15.11	12.88
Ours (PWSV)	**25.82**	14.96	18.36
Ours (GPLL)	25.31	**31.55**	**18.55**

**Table 3 entropy-26-00328-t003:** Evaluation of the segmentation results ODS, OIS, and AP of different methods on the DeepCrack dataset (%).

Methods	ODS	OIS	AP
FSV	46.43	54.97	30.95
CAM [46]	44.88	52.43	37.33
PWSV [13]	37.05	43.95	44.31
GPLL [12]	65.97	73.19	72.28
Ours (CAM)	49.66	53.18	47.22
Ours (PWSV)	69.47	63.31	73.91
Ours (GPLL)	**71.01**	**77.98**	**75.51**

**Table 4 entropy-26-00328-t004:** Performance of different modules in the model on Crack500 testing dataset (%).

Co-Training	Dynamical Division	ODS	OIS	AP
		45.04	56.69	45.46
✓		56.86	63.74	59.09
✓	✓	**60.43**	**64.50**	**63.65**

## Data Availability

The data presented in this study are available on request from the corresponding author. Implementation of the proposed framework is publicly available on GitHub at the following link: https://github.com/liangfengjiao/CAC.

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
