# Peer review of "CAC: Confidence-Aware Co-Training for Weakly Supervised Crack Segmentation"

_entropy, 2024, doi:10.3390/e26040328_

Round 1
Reviewer 1 Report
Comments and Suggestions for Authors
General remarks :
1) This paper addresses a pixel-wise segmentation of cracks in presence of inaccurate labels. This is a real challenge and such as gives this paper some importance.
2) A weak supervision itself is also a hot topic and joint training of two networks on high-confidence and low-confidence labels is indeed novel.
Unfortunately there are a few issues with this paper that need to be properly addressed before this paper could be considered for publication.
Major comments :
My principal concern is related to that the paper pretends to accurately segment cracks. Provided ground-truth labels are supposedly noisy. The results are nonetheless scored against this noisy GT. This is a problem for me. Segmentation of cracks in presence of noisy labels was also addressed in [1, 2]. These papers should be cited. In [2] the results were also scored using unsupervised scores, and also on a sythetic dataset allowing to control the amount of noise on the labels. I would expect same kind of evaluation here.
Choice of thresholds \tau_1$ and $\tau_2$ , and their sensitivity is not justified.
section 4.2 Evaluation metrics - Given that in all figures (e.g. Fig 4) your results are in grey, how do you obtain binary result ? What is the meaning of Pr, Re and F1 is GT is assumed noisy?
Minor comments :
line 223 : "This approach aims to maximize the activation of the crack area within the entire image." It is not clear why?
line 234 : M is not binary
eq 1. what are ij?
line 257: $\alpha$ is undefined
line 276: independent parameters, $\theta$ and $\theta'$ ... they do not seem independent because later, Fig 2, \theta' is updated using \theta. Which is , BTW, difficult to imagine. This needs to be clarified.
line 281: statement "are masked to hide the low-confidence cracked pixels portion" needs to be clarified.
equation 13. Given the integration sum limit T is 100, should the percentile 0.01 be 1?
line 428: Statement "The CAC demonstrated greater resistance to noise interference compared to other methods" needs justification.
A few typos are present here and there in the text and should be corrected.
[1] https://doi.org/10.1016/j.neucom.2022.01.051
[2] https://hal.science/hal-03451685
quality acceptable after correcting a few typos
Reviewer 2 Report
Comments and Suggestions for Authors
In this work, an interesting confidence-aware co-training (CAC) framework is proposed for weakly supervised crack segmentation (WSCS). Extensive experiments conducted on the Crack500, DeepCrack, and CFD datasets demonstrate that the proposed CAC significantly outperforms other WSCS methods.
In the ablation study part, the authors need to conduct more detailed experimental instructions on the parameter selection of the proposed method.
Comments on the Quality of English LanguageThe writing quality of the English needs to be greatly improved. There are many obvious grammatical and spelling errors. I hope that the improved version received next time will have a big improvement on the English writing of the article. Please mark the changed parts in other colors.
Round 2
Reviewer 1 Report
Comments and Suggestions for Authors
After reading the revised version of the paper I am still having some issues. Although the authors addressed my remarks some of them still need clarification.
The english writing has indeed improved.
Major comments :
I am still having same issues as I have already pointed out previously to which the authors failed to answer satisfactorily :
Training: I still can not understand how two networks can be co-trained so that both receive gradient descent from only one, the $\Nabla\theta$. The authors refer to MAML [Finn et al, 2017] paper, but for me the context is different. In this paper, there is no such collection of similar tasks as in MAML and a few-shot fine-tuning. This all needs a clarification. How both networks are initialized, pretrained, and why does the co-training converge towards optimally detected cracks.
Evaluation: The GT of the crack datasets used in this paper can in no way be considered as accurate. There are apparent errors (missing cracks) in GT already in Fig. 4 (b). Your results seem to converge towards a solution with same cracks missing as those in the GT. Consequently, the evaluation against an inaccurate GT is pointless. What is the meaning of the numbers in Tab. 1? A higher score only means a closer result to an inaccurate GT. For me a sound evaluation could be using a synthetic crack dataset where a GT is accurate, and where you could control the amount of noise on the labels, and observe the influence this amount of noise has on your result. There are a few available, e.g. syncrack our supervisely. A sound evaluation would definitely increase the impact of your paper.
Minor (or typo) comments :
Figure 1. Segmentaion
line 200: implicitly mine ???
line 235: misleading, instead of ... "P, represented as a gray-scale image, is converted into a gray-scale image M via threshold segmentation" prefer saying something like : "pixels under t\tau_1 are set to zero"
line 260: replace "dimension" by "size"
line 286: replace cracked by crack
